# The Use of Thermography as an Auxiliary Method for Monitoring Convalescence after Facelift Surgery: A Case Study

**DOI:** 10.3390/ijerph19063687

**Published:** 2022-03-20

**Authors:** Monika Chudecka, Andrzej Dmytrzak, Katarzyna Leźnicka, Anna Lubkowska

**Affiliations:** 1Institute of Physical Culture Sciences, Faculty of Physical Education and Health, University of Szczecin, 71-065 Szczecin, Poland; monikachudecka@wp.pl; 2Aesthetic Med, 71-403 Szczecin, Poland; aestheticmed@aestheticmed.com.pl; 3Faculty of Physical Culture, Gdansk University of Physical Education and Sport, 80-336 Gdansk, Poland; k.leznicka@tlen.pl; 4Department of Functional Diagnostics and Physical Medicine, Faculty of Health Sciences, Pomeranian Medical University, 71-210 Szczecin, Poland

**Keywords:** thermography, facelifting, rhytidectomy, face temperature

## Abstract

Although IR thermography is widely used in medical diagnostics, there are no reports that describe the use of IR thermography in the evaluation of post-plastic-surgery regeneration processes. The aim of the study was to evaluate the potential of thermography as a method which, among others, allows us to determine the location and extent of the inflammatory process, supporting the clinical evaluation of the patient’s convalescence after a facelift surgery using the SMAS technique. During the study and in order to monitor the convalescence process, the patient had a series of face thermograms performed before surgery and up to the 6th week after it. The healing process after surgery was multidirectional for the contralateral areas of the face, leading to thermal asymmetry lasting up to the 3rd week of convalescence. The lowest T_mean_ values for ROIs were recorded in week 3 of the study and then they gradually increased, in week 6 after surgery, to the following values: chin = 33.1 ± 0.72 °C; cheek left = 33.0 ± 0.26 °C; cheek right = 33.2 ± 0.51 °C; ZFL = 33.8 ± 0.45 °C; ZFR = 33.6 ± 0.74 °C; ZLL = 32.6 ±0.55 °C; ZLR = 32.3 ± 0.32 °C. The temperatures of these areas were still lower than the baseline values obtained before surgery by 0.5–1.4 °C. The usefulness of thermography in the evaluation of post-operative convalescence in facial plastic surgery procedures shows potential in the context of diagnostic assessment of the dynamics of changes in the healing process.

## 1. Introduction

During past few years, there was a noticeable come back of studies on the human face aging, face attractiveness, face recognition as well as facial skin rejuvenation. Facial aging is a dynamic process involving the aging of soft tissue, including the laxity of the overlying skin and bony structures. This complex process involves two important factors: fat atrophy, loss of elasticity, volume loss throughout the face, and repetitive muscle movements that cause wrinkles and folds [1]. Treating the aged face requires an understanding of bone and soft tissue anatomy, including the analogous lamellar layers of the face and neck, and the techniques designed to restore youthful skin tone and facial contours. Although volume restoration with fillers is effective for restoring youthful facial contours, the power of face lifting is unmatched in its ability to rejuvenate a sagging facial shape [2].

Traditionally, facial rejuvenation has focused on various dermatologic noninvasive and invasive cosmetic procedures, e.g., carbon dioxide, laser resurfacing, microdermabrasion, injection of mesotherapy products, electric stimulation leading to collagen production in human skin fibroblasts. Several procedures, predominantly facelifts, can be performed to smooth wrinkles, strengthen ligaments, and reposition fat pads to reverse age-related changes and confer a more youthful appearance. The goal of facelift surgery is to rejuvenate and improve the appearance of the face. A lift can improve volume shifts with age, reposition sagging tissues, and eliminate skin redundancy [3].

Currently, rhytidectomy supplemented with superficial muscular aponeurotic system (SMAS) technique is considered the most effective of invasive rejuvenating procedures aimed at removing excess skin and adipose tissue and restoring the tone of the myofascial layer [4]. SMAS is tethered to the deep fascia by several retaining ligaments, including zygomatic ligaments in the inferior border of the zygoma, masseteric ligaments along the anterior edge of the masseter muscle, the platysma auricular ligament over the parotid gland, and the mandibular ligament from the parasymphysial mandible. Between fixed regions, SMAS slides freely on the deep fascia [5,6].

It is accepted today that, regardless of the technique used, any facelift procedure should consider the fact that the deeper tissues must be repositioned or filled before the skin is pulled and resected [7]. However, surgical procedures are associated with the possibility of complications such as infection, skin necrosis, haematomas, ecchymoses, damage to the anterior and marginal branches of the facial nerve, as well as the risk related to general anaesthesia or even deliberate sedation. They are also associated with visible scarring and long-term recovery [8]. Apart from nerve injuries or haematomas, one of the main risks associated with face lifting is the development of skin necrosis, disturbance of facial skin microcirculation, vascular damage, seroma (i.e., fluid accumulating between the skin and deeper tissues), or lymphoedemas.

Many diagnoses of diseases that affect the facial region are made by imaging methods. A method that is increasingly used in clinical medical diagnostics and which can also illustrate the changes described during postoperative face regeneration could be thermography taking into account, based on the measurement of skin surface temperature, functional aspects, such as regional microcirculation, inflammatory processes and the autonomous nervous system.

In order to maintain normal thermoregulation, the neurovegetative central nervous system, by means of the hypothalamus, controls the skin blood flow in a uniform and symmetrical manner, resulting in a right/left thermal pattern that is also symmetrical [9]. However, the qualitative and quantitative changes in thermal distribution are indicative of abnormality. Furthermore, the examination allows greater precision in thermal quantification and monitoring of the region of interest (ROI) [10].

### Aim of the Study

To the best of our knowledge, there are no reports that describe the use of IR thermography in the evaluation of post-plastic surgery regeneration processes. During facial surgery, the continuity of skin tissues, subcutaneous fat, small nerve branches, lymph vessels, blood vessels, fasciae and muscle fibres is broken. The tiny blood and lymph vessels are damaged and the body triggers an inflammatory response. The analysis of thermograms can, therefore, be used to select areas in which the inflammatory process occurs (increased temperature in relation to symmetrical or adjacent areas) or areas with less blood supply as a result of an invasive surgical procedure (reduced temperature). The temperature distribution and area temperature variation may be abnormal due to changes in blood flow. Analysis of temperature changes within the same areas in a series of examinations can be used to monitor the healing process. The aim of this study is, therefore, to assess the potential of thermography as a method that, among others, allows us to determine the location and extent of the inflammatory process, supporting the clinical evaluation of the patient’s convalescence after facelift surgery. The paper presents a case of a female patient in whom the convalescence process after a facelift surgery was monitored with the use of a thermal imaging camera.

## 2. Materials and Methods

The study was conducted at the clinic Aesthetic Med Andrzej Dmytrzak, Centre of Plastic and Reconstructive Surgery in Szczecin (Poland). The research was approved by the Bioethics Committee of the Pomeranian Medical University (Ref. No. 10/KB/VI/2018). The study was performed in compliance with the Declaration of Helsinki.

### 2.1. Description of the Patient

A 45-year-old female patient (BMI = 20 kg/m^2^), Caucasian, in general good health, qualified for facial plastic surgery with soft silicone implants on the shaft of the zygomatic bone after medical consultation. The factor accelerating the involutional changes in the patient’s face was the suspicion of Ehlers-Danlos syndrome (EDS), genetically burdening the patient with abnormal structure of basic components of connective tissue and metabolic disorders within it. It is characterised, among others, by hyperelastic skin, hypermobile joints and poor and slow healing of wounds. Collagen loss progresses with age and accelerates the process of bone loss, osteoporosis, and contributes to premature aging [11].

The indications for the facelift surgery in the patient were

·Skin laxity in the medial part of the face and the jawline;·Deep nasolabial folds;·Wrinkles coming from the corners of the mouth down the chin (marionette lines);·Loss of facial fat (volume) in the area ofthe zygomatic bone that has been filled with the implant;·Sagging and loss of muscle tone in the lower part of the face, causing the facial contours to fall;·In general, excessive stretching of the facial skin in the medial and lower regions.

### 2.2. Methodology of the Performed SMAS Procedure

The SMAS facelift was performed—an invasive procedure aimed at removing excess skin and restoring the tension of the myofascial layer in the medial and lower parts of the face. Under general endotracheal anaesthesia, after preparation of the operating field, an incision was made in the temporal, pre-auricular and post-auricular areas. Additionally, an incision was made along the lower edge of the orbitalis oculi muscle, a subperiosteal pocket was created along the zygomatic bone—haemostasis. A zygomatic implant was inserted into the pocket. An incision was made in the facial fascia along the zygomatic bone, the incision was extended vertically in the pre-auricular area—3 cm from the auricle. The incision was made to the angle of the mandible. The fascial flap was detached from the substrate. The produced flap was pulled upwards, obtaining its excess of 2 cm in length and 2.5 cm in width. The flap was anchored to the substrate. Sutures were applied.

Drainage tubes were also installed to drain blood, body fluids and lymph. An absorbent dressing and a tourniquet were used. The surgery lasted 6 h, the patient remained under general anaesthesia. Hospitalisation lasted 2 days.

In the first few days, the face was swollen and bruised. The swelling lasted for about two weeks.

### 2.3. Thermography

During the study and in order to monitor the convalescence process, the patient had a series of face thermograms performed in the following study series:Series 1—1 day before facelift surgery;Series 2—3 days after facelift surgery;Series 3—5 days after facelift surgery—after removing the sutures under the eyes;Series 4—8 days after facelift surgery;Series 5—11 days after facelift surgery—after removing all the sutures;Series 6—3 weeks after facelift surgery;Series 7—1 month after facelift surgery;Series 8—6 weeks after facelift surgery.

Face surface temperature measurements were performed in each case in the morning (before 11 a.m.). The patient was subjected to thermography in a standing position using a FLIR T1030sc HD camera (Teledyne FLIR LLC 27700 SW Parkway Avenue Wilsonville, OR 97070 USA), each time after 20 min of acclimation. T1030sc HD camera is a high-speed imaging and measurement camera that records 1024 × 768 HD resolution images (pixels) at 30 frames per second with accuracy ±1 °C. Spectral range: 7.5–14 μm. Digital thermograms were carried out on the front of the face and also for the left (L) and right (R) side of the subject’s face, each time specifying the area of the forehead (F), the area of the zygomatic bone frontal (ZF), zygomatic bone lateral (ZL), the cheek area (C), the nose area (N), the chin area (Ch) and medial palpebral commissure (MPC), i.e., the place of the anatomical surfaces of the body that most accurately reflects the internal temperature of the body [10].

The camera was positioned in a straight line to the subject, 1.5 m from the face. The measuring procedures were carried out following the standards of the European Thermographic Association [12]. Standardised conditions of the examination room are extremely important for performing the thermographic examination. In the room, a thermometer was available during all the examinations, and it was positioned in a place away from any heat sources in it, in order to monitor it. The temperature variation in the room did not exceed 0.5 °C. During the imaging process, the ambient temperature and humidity were constant at the measurement site, i.e., 25 °C ± 0.5 °C and 55–60%, respectively. Skin emissivity was adopted as 0.98. At the same time during IR examination, the right and left tympanic temperatures were measured. Figure 1 shows the regions of interest (ROIs), determined in the FLIR ResearchIR software (Teledyne FLIR LLC 27700 SW Parkway Avenue Wilsonville, OR 97070 USA), using the functions of graphical determination of the analysis area in the form of templates, used consistently in subsequent thermogram analyses. For each selected face surface area, the mean surface temperature (Tmean) with standard deviation was calculated.

## 3. Results

The baseline internal body temperature of the patient, as reflected in the tympanic measurements, was 36.5 °C in the right and left ear. The numerical values obtained from measurements before facelift surgery for individual ROIs are summarised in Table 1.

The pre-surgery facial temperatures ranged from the lowest values for the nose (T_mean_ = 33.5 ± 0.79 °C) to the highest at MPCR and MPCL measurement points (T_mean_ = 36.4 °C and T_mean_ = 36.3 °C, respectively), which were comparable to the values of tympanic temperatures. At the same time, no thermal asymmetry was observed between the right and left sides of the face.

The mean values of the patient’s ROI temperatures in the series of examinations after facelift surgery are summarised in Table 2. In the subsequent series of examinations after facelift surgery, the thermal analysis showed that relatively constant values for the tympanic temperature (36.1–36.9 °C) and for the MPCR and MPCL measurement points (36.3–36.6 °C) throughout the observation period, i.e., for the period of 6 weeks after facelift surgery, were maintained.

On the 3rd day after surgery, a much larger thermal gradient between the analysed areas of the face was recorded, from the lowest values for the nose area (T_mean_ = 26.6 ± 1.20 °C) to the highest ones in areas on the left side of the face: cheek (T_mean_ = 33.8 ± 0.62 °C), ZFL (T_mean_ = 33.9 ± 0.62 °C) and ZLL (T_mean_ = 33.7 ± 0.33 °C). On this day, a significantly lower temperature was recorded in relation to the preoperative values for the following areas: nose (T_mean_ = 26.6 ± 1.20 °C vs. 33.5 ± 0.79 °C), forehead (T_mean_ = 33.4 ± 0.34 °C vs. 34.7 ± 0.42 °C), ZFL (T_mean_ = 33.9 ± 0.62 °C vs. 34.3 ± 0.79 °C) and ZFR (T_mean_ = 33.4 ± 0.64 °C vs. 34.1 ± 0.77 °C), chin (T_mean_ = 33.1 ± 0.82 °C vs. 34.3 ± 0.75 °C), ZLL (T_mean_ = 33.7 ± 0.33 °C vs. 33.9 ± 0.30 °C) and ZLR (T_mean_ = 33.2 ± 0.62 °C vs. 33.7 ± 0.51 °C), cheek left (T_mean_ = 33.8 ± 0.62 °C vs. 34 ± 0.46 °C) and right (T_mean_ = 33.6 ± 0.15 °C vs. 34 ± 0.54 °C). Comparing symmetrical areas of the face, a thermal asymmetry was demonstrated for the zygomatic area, both in the frontal and lateral views, amounting to 0.5 °C in both cases.

The sutures were removed in the infraorbital region. After that, on the 5th day from the surgery, an increase in temperature compared to the 3rd day in the areas of the forehead (33.8 ± 0.81 °C) and nose (32.8 ± 0.63 °C) was observed, but it was still lower compared to the baseline temperatures before surgery. In other analysed areas, temperatures were lower compared to the measurement series, both the first series (before surgery) and the second one (3 days after surgery). Comparing the symmetrical areas of the face, a thermal asymmetry for the zygomatic areas was again demonstrated. both the frontal and lateral views, with the same thermal differentiation as on day 3 (0.5 °C).

During the next scheduled measurement, on the 8th day after surgery, a further increase in temperatures was observed in the area of forehead (34.3 ± 0.62 °C), ZFL (34.1 ± 0.72 °C) and ZFR (33.3 ± 0.90 °C), and chin (33.7 ± 0.82 °C), still being lower than the baseline T_mean_ from the time before surgery. Simultaneously, a decrease in T_mean_ was recorded, by 3.7 °C, in the nose area vs. the 5th day after surgery. In other areas, in the lateral views, changes in temperature on that day were multidirectional. On the left side of the face there was a slight increase in T_mean_ for ZLL and cheek compared to both 3rd and 5th day after surgery, while on the right side of the face an increase was observed for the ZLR area compared to 5th day and a slight decrease in T_mean_ for the cheek area both compared to both 3rd and 5th day after surgery.

In addition, a greater thermal asymmetry was marked for the zygomatic areas, reaching 0.8 °C both in the frontal and lateral views. For the cheek areas, a thermal asymmetry was demonstrated at 0.7 °C. Ten days after the patient’s surgery, all sutures were removed and subsequent thermal measurements, performed on day 11, showed a persistent temperature for the forehead (T_mean_ = 34.3 ± 0.44 °C and nose (T_mean_ = 29.2 ± 1.00 °C) areas. Temperature in other areas analysed in the frontal view increased compared to earlier measurements, obtaining the following values: T_mean_ = 34.3 ± 0.63 °C for ZFL, T_mean_ = 34 ± 0.63 °C for ZFR, and T_mean_ = 34.7 ± 0.47 °C for the chin area. Interestingly, on this measurement day, the direction of changes in the temperature values of the right and left side of the face led to the disappearance of thermal asymmetry, owing to the increase in temperature on the right side and the decrease on the left side compared to 8th day of thermal analyses. From the 3rd to 6th week after surgery, the temperature of the forehead area was maintained on a relatively constant level, obtaining subsequently 34.5 ± 0.46 °C, 34.5 ± 0.47 °C and 34.6 ± 0.36 °C, being comparable to the baseline values for this area, equal to 34.7 ± 0.42 °C. The temperature of the nose area increased from 30.4 ± 0.28 °C in the 3rd week after surgery, up to 31.4 ± 1.00 °C in 4th week and 32.6 ± 0.90 °C in the 6th week after surgery, still being lower than the baseline temperature by 0.9 °C. The areas of chin, cheek and zygomatic in the frontal and lateral views were characterised by the lowest temperature values in the 3rd week after surgery compared to all measurement days, but without thermal asymmetry for contralateral sides. The lack of thermal asymmetry remained to the end of the analysis, i.e., up to week 6th after surgery. From 3rd week after surgery, the temperatures of these areas gradually increased, reached the following values of T_mean_: chin = 33.1 ± 0.72 °C; cheek left = 33 ± 0.26 °C; cheek right = 33.2 ± 0.51 °C; ZFL = 33.8 ± 0.45 °C; ZFR = 33.6 ± 0.74 °C; ZLL = 32.6 ± 0.55 °C; ZLR = 32.3 ± 0.32 °C in the 6th week after surgery. The temperatures of these areas remained lower than the baseline values obtained before the surgery performed by 0.5–1.4 °C. (See Figure 2).

## 4. Discussion

The facial skin temperature is an important parameter which gives useful information about thermal comfort sensation, metabolic activity of tissues and pain that appeared to be associated with inflammation. The infrared thermography is known as an excellent tool for visualising soft tissue infections in humans and can provide a physiological indicator of the underlying disease [13]. A change in temperature in the affected regions, as one of the main characteristics of inflammation, assessed using the thermal imaging method, allows considering this method as diagnostically useful in identifying and localising inflammation.

The main aim of the presented case study was to check if IR thermography could be useful in controlling the course of the healing of the facial area and assessing the maintenance of thermal symmetry of the contralateral sides in the facial area after the invasive SMAS facelift surgery. In order to maintain proper thermoregulation, the vegetative nervous system uses the hypothalamus to control the blood flow in the skin in a uniform and symmetrical way, resulting in a symmetrical right/left thermal pattern. Consequently, the temperature in the area of the face, as well as the temperature of the entire body, is symmetrical in healthy people under conditions of thermal comfort. The thermographic image of the face of healthy people is characterised by symmetry (temperature difference between the left and right side ∆t < 0.3 °C), cold nose, warm eye sockets, no hot foci in the projection of the maxillary sinuses [14].

The first studies of facial temperature with the use of thermography concerned thermal changes as psychophysiological symptoms accompanying emotions. Importantly, thermography created the possibility of a comprehensive examination, with a temperature map, not a unit measurement. Changes in the surface temperature of the face under the influence of emotions and subjective reactions of the body and at the same time the sensitivity of this method in assessing changes in the temperature of the face area were confirmed in the studies by Polakowski et al. and by Wu et al., extending the analysis to include the novel method for infrared face recognition based on blood perfusion [15,16]. Rapiejko et al. confirmed in clinical trials that face tomography may be a non-invasive method of screening assessment of the condition of the nose and paranasal sinuses and may be a supplement to the routine diagnosis of diseases in these areas of the body [17].

It is worth noting that in the case of assessing the psychosomatic response, changes in the temperature of the face surface by 0.2 °C on average were considered significant, which is a relatively small value, taking into account the fact that in the assessment of thermal symmetry distribution of the human body, 0.5 °C is taken as the normative allowable difference related to lateralisation. In 2012, Vardasca et al. determined precisely that the thermal lateral differentiation in healthy people is 0.4 ± 0.3 °C when the entire body surface is considered and 0.4 ± 0.15 °C for its individual regions [18].

Due to this regularity concerning high thermal symmetry, thermography seems to be a useful tool in studies and clinical diagnostics of the facial area and, in our opinion, also in the context of the postoperative healing process [19]. The obtained results concerning the temperatures of the analysed surfaces are compared with the accepted normative ranges, or symmetrically to the opposite side of the body. The qualitative and quantitative changes in thermal distribution are indicative of abnormality. These asymmetrical patterns usually occur due to the presence of sympathetic changes, traumatic lesions, or vascular or inflammatory changes—which was the case of the patient treated in this study [20].

The analysis of the thermograms made in the patient before surgery confirmed the symmetrical distribution of facial temperatures. At the same time, they confirmed the thermal differentiation of the analysed areas, while maintaining thermal comfort during taking thermographic images. It can be concluded that the temperature range for the face is in the range from 33.5 ± 0.79 °C to 34.7 ± 0.42 °C. As expected and in line with the literature data, the nose was the coldest area [21]. The skin temperature obtained from the thermal images is related to the blood flow through the skin, controlled by the sympathetic part of the autonomic nervous system. The distribution and changes in skin temperature depend on the conduction of heat from the skin’s blood [22]. In particular, the temperature of the facial skin shows a capacitive variation that reflects the dermal-vascular capacity and the heat capacity of the cutaneous tissue. This capacitive variation is spatially dependent on differences in the arrangement of blood vessels and the structure of the cutaneous tissue in the facial skin [23]. Various articles have evaluated thermographic quantitative changes in facial skin vasomotor neurovegetative activity, that is, they calculated the absolute temperature of the ROI and conjugated gradients [19,24]. When analysing the hot and cold areas of the face, the literature is unanimous about dividing them in the following manner: hot areas—frontal, orbital, labial, nasolabial, auricular appendages, temporal; and cold areas—cheeks, hair and hairy areas, auricular pavilion or auricle, mento/chin [10].

Liu et al. suggested that cardiovascular activities presented different dynamic properties within different facial sub-regions; moreover, blood vessels in the nasal region pass through a small space between the nasal bone and the skin. Furthermore, the nasal region has a higher concentration of blood vessels than other parts of the face [25].

When analysing the temperature distribution within the face before surgery, it was also shown that the thermal values in the point measurement for the medial palpebral commissure region, both for right and left side, are comparable with the body core temperature, with T_mean_ = 36.4 °C and 36.3 °C for MPCR and MPCL, respectively. It has been proved that inner canthi (palpebral commissures) are considered to be optimal locations for non-contact temperature measurement. Perfused by the internal carotid artery, they are typically the warmest regions on the face and have high stability and strong correlation with internal body temperature [26,27,28,29].

The worst moment for a facelift patient is the postoperative recovery. Pain, swelling, and bruising are the main reasons. As a result of every surgery, dead cells, clots, blood residue, and alteration of the blood supply leads to postoperative oedema. During a rhytidectomy, the incision and dissection of the skin cause interruption of the normal venous and lymphatic drainage. Since the medial parts of the face and neck are the base of the pedicle of this often-large flap, the venous and lymphatic flow of the face must revert its previous course towards the parotid and submaxillar lymphatic nodes, to the medial area of the face. This abrupt reversal and reconversion of the flow provokes, in part, the oedema of the flap [30].

When analysing the changes in temperatures after the facelift surgery, a decrease was observed compared to the baseline values. Subsequently, on day 11 of convalescence (the day after removing the sutures), there was a significant increase in temperatures for selected areas, i.e., zygomatic ZFL and ZFR and chin, as well as the right side ZLR and cheek, reaching the highest values of all series of examinations performed after surgery, but not exceeding the baseline temperatures. In addition, the analysis of thermograms made in the following days of convalescence showed that thermal changes in symmetrical areas of the face were not unidirectional, simultaneously decreases and increases in temperature in contralateral ROIs were observed, which may indicate a different intensity of the healing process in these areas. The variability of the thermal response persisted up to 3 weeks after procedure. As a consequence, after the surgery, thermal asymmetry of the face appeared, concerning the zygomatic L&R and cheek L&R areas, which also persisted until the third week after surgery. The causes of the occurrence of thermal asymmetry can be seen in the various postoperative activity of the vessels under the skin resulting from increased or decreased local blood flow around the operated sites, disturbances in filtration and reabsorption, and the inevitable surgical interference in the lymphatic system vessels, leading to oedema.

It seems that the third week of convalescence of the examined patient was a breakthrough for the thermal response of the operated area. In this examination series, the patient had the lowest temperatures of ROIs (except forehead and nose) recorded of all the examination series. The nasal region relates to sympathetic nervous system activity because the nasal region has a high concentration of arteriovenous anastomosis (AVA) regulating the blood flow volume. On the other hand, changes in forehead skin temperature have less autonomic nervous activity than the nasal region because of having less vasoconstriction action caused by vasoactive nerves [31].

Moreover, the Tmean value for contralateral ROIs was found not to exceed 0.4 °C. It can therefore be concluded that in the examined patient, the return of thermal symmetry of the face occurred in the third week after surgery and it was maintained until the end of our study. The last thermographic assessment of ROIs was performed 6 weeks after surgery, when the patient was allowed us to return to regular activity [32]. It should be noted, however, that in the presented study, the values of temperatures for the patient’s ROIs (Tmean)did not return to the baseline temperatures.

Based on the thermographic examinations carried out in the female patient undergoing a facelift surgery, it can be concluded that the nature of changes in the analysed temperatures (maintained lower than the baseline T_mean_ of ROIs in subsequent examination series) may indicate that in the case of proper tissue healing, the main factor determining changes in the temperature of the face surface after facelift surgery is disturbances in the peripheral circulation of the operated area, not postoperative inflammation. However, the observed transient slight increase in the T_mean_ of ROIs at day 11 compared to other stages of convalescence may indicate an increase in inflammation associated with a response to tissue discontinuity during this period. After the skin is lifted, the drainage flow to the flaps is reversed abruptly toward the medial part of the face, where the flap bases are located. The thickness and extension of the flap determines the magnitude of the postoperative oedema, which is also augmented by medial surgeries (blepharo, rhino) whose trauma obstruct their natural drainage, increasing the congestion and oedema [30]. Inflammatory processes change the image of the heat flux, which results in a large temperature gradient between the pathologically changed area and its immediate surroundings. The temperature values on the skin surface and the rate of their changes, which, thanks to the thermal imaging camera, can be recorded and archived with a resolution of less than ±1 °C, may therefore be a resultant of postoperative changes in circulation and the inflammatory response. Inflammatory processes appear in thermograms as areas of increased temperature, while hypoperfusion is seen as areas of lower temperatures.

## 5. Conclusions

The values of temperatures for ROIs (T_mean_) analysed in the examination series within 6 weeks after surgery did not return to the patient’s baseline temperature values. In the third week after surgery, the lowest temperatures of ROIs were recorded among all the examination series. As a result of the facelift surgery, a thermal asymmetry of the face appeared, which persisted for up to week 3 after surgery. The healing process was non-simultaneous for contralateral areas, and the values of temperatures for ROIs were the resultant of the inflammatory response of tissues and changes in the subcutaneous circulation. The usefulness of thermography in the assessment of postoperative convalescence in facial plastic surgery procedures shows potential in the context of diagnostic assessment of the dynamics of changes in the healing process.

## Figures and Tables

**Figure 1 ijerph-19-03687-f001:**
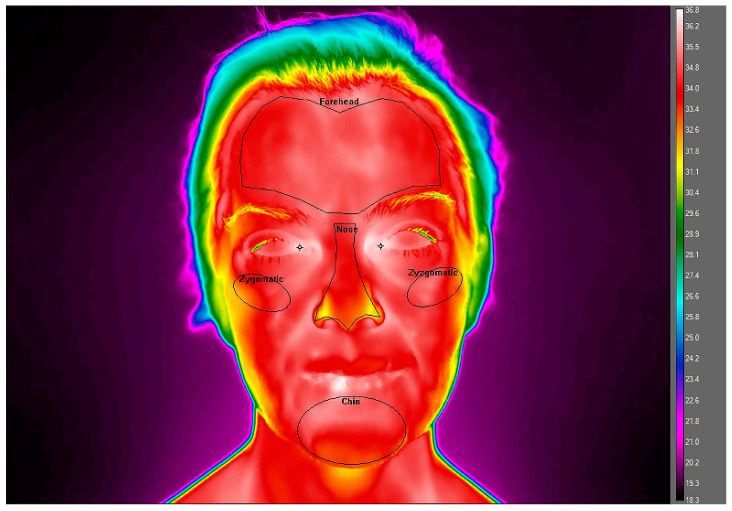
Photo of the patient before surgery. The regions of interest (ROIs: face front, left and right side) selected for thermal analysis were marked.

**Figure 2 ijerph-19-03687-f002:**
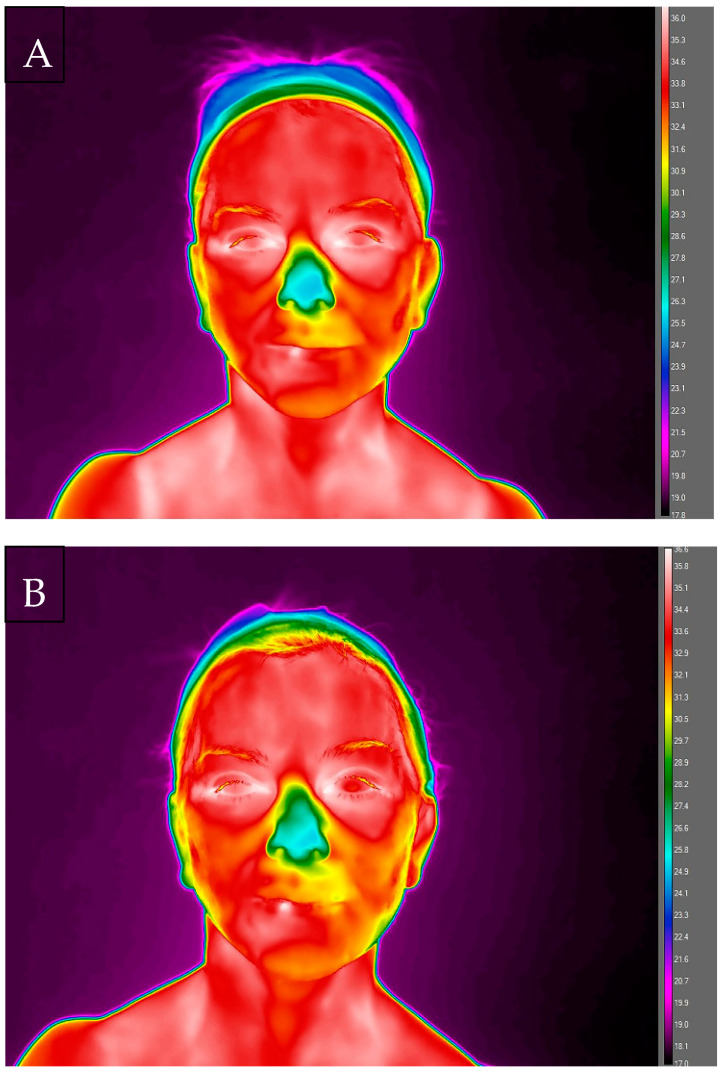
Thermograms of the patient’s face in the subsequent series of examinations in the frontal view: (**A**)—3 days after facelift; (**B**)—5 days after facelift; (**C**)—11 after facelift; (**D**)—1 month after facelift.

**Table 1 ijerph-19-03687-t001:** Mean values of patient ROI temperatures before facelift surgery, obtained in thermographic examinations.

ROIs	T_mean_ °C	±Sd
Front		
Forehead (F)	34.7	0.42
Nose (N)	33.5	0.79
Zygomatic (ZFL)	34.3	0.79
Zygomatic (ZFR)	34.1	0.77
Chin (Ch)	34.3	0.75
MPC L	36.3	
MPC R	36.4	
Left side		
Zygomatic (ZLL)	33.9	0.30
Cheek (C)	34	0.46
Right side		
Zygomatic (ZLR)	33.7	0.51
Cheek (C)	34	0.54
Tympanic		
right		left
36.7		36.6

**Table 2 ijerph-19-03687-t002:** Mean values of patient ROIs temperatures (T_mean_) in the series of examinations after facelift surgery.

	3 Days after Surgery		5 Days after Surgery		8 Days after Surgery		11 Days after Sugery		3 Weeks after Surgery		1 Month after Surgery		6 Weeks after Surgery	
	T_mean_	±Sd	T_mean_	±Sd	T_mean_	±Sd	T_mean_	±Sd	T_mean_	±Sd	T_mean_	±Sd	T_mean_	±Sd
*Front*														
Forehead	33.4	0.34	33.8	0.81	34.3	0.62	34.3	0.44	34.5	0.46	34.5	0.47	34.6	0.36
Nose	26.6	1.20	32.8	0.63	29.1	0.91	29.2	1.00	30.4	0.28	31.4	1.00	32.6	0.90
Zygomatic ZFL	33.9	0.62	33.2	0.81	34.1	0.72	34.3	0.63	31.9	0.80	33.4	0.66	33.8	0.45
Zygomatic ZFR	33.4	0.64	32.7	0.71	33.3	0.91	34	0.63	31.7	0.91	33.5	0.74	33.6	0.74
Chin	33.1	0.82	32.8	0.91	33.7	0.82	34.7	0.47	31.8	0.90	33.0	0.74	33.1	0.72
MPC L	36.5		36.3		36.4		36.5		36.2		36.4		36.4	
MPC R	36.5		36.4		36.3		36.3		36.3		36.3		36.4	
Left side														
Zygomatic ZLL	33.7	0.33	33.1	0.45	34.0	0.54	33.5	0.25	31.9	0.66	32.4	0.44	32.6	0.55
Cheek	33.8	0.62	33.2	0.71	34.0	0.63	33.9	0.55	31.6	1.00	32.1	0.43	33.0	0.26
Right side														
Zygomatic ZLR	33.2	0.62	32.6	0.61	33.2	0.80	33.4	0.64	31.5	0.48	32.2	0.37	32.3	0.32
Cheek	33.6	0.15	33.5	0.43	33.1	0.56	33.8	0.57	31.8	0.81	32.5	0.54	33.2	0.51
Tympanic														
right	36.9		36.7		36.4		36.6		36.1		36.3		36.5	
left	36.7		36.6		36.5		36.5		36.2		36.3		36.4	

## Data Availability

Not applicable.

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
