# Peer review of "The Use of Thermography as an Auxiliary Method for Monitoring Convalescence after Facelift Surgery: A Case Study"

_ijerph, 2022, doi:10.3390/ijerph19063687_

Round 1
Reviewer 1 Report
The manuscript provides original data on the use of thermography as an auxiliary method for monitoring convalescence after facelift surgery. The paper is potentially of interest and suitable for International Journal of Environmental Research and Public Health. The issue is an “hot topic”;the study is of interest and worth of publication.However, a few points should be clarified before publishing:- details should be given regarding the technical parameters of the camera, which may be important for the sensitivity and the possibility of using thermography in post-surgical regeneration studies in the face;
- please explain what it means in Tab. 2 in the second column of results in brackets „(sutures)”. If it is explained in the text there is no need to include it in the table
- The authors report that the right and left tympanic temperatures were measured, at the same time during IR examination, but these results have not been presented in the study.
Author Response
We would like to thank you very much for your valuable comments. We have made changes taking into account the reviewer's suggestion. All changes in the text are marked in red
Reviewer comments
The manuscript provides original data on the use of thermography as an auxiliary method for monitoring convalescence after facelift surgery. The paper is potentially of interest and suitable for International Journal of Environmental Research and Public Health. The issue is an “hot topic”;the study is of interest and worth of publication.However, a few points should be clarified before publishing:
- details should be given regarding the technical parameters of the camera, which may be important for the sensitivity and the possibility of using thermography in post-surgical regeneration studies in the face; details regarding the technical parameters of the camera have been added
- please explain what it means in Tab. 2 in the second column of results in brackets „(sutures)”. If it is explained in the text there is no need to include it in the table.
The term "sutures" referred to the day of the examination on which the sutures were removed. Due to the explanation in the text, it was removed from the table in accordance with the reviewer's comments.
- The authors report that the right and left tympanic temperatures were measured, at the same time during IR examination, but these results have not been presented in the study. The initial results and the results from the following days should be given in the temperature tables and not just a scope throughout the study.
The values ​​of tympanic temperature have been added.
Reviewer 2 Report
The article presents the use of IR thermography in the evaluation of postoperative regeneration processes. The article examines the use of thermography as a method supporting clinical evaluation of the patient's convalescence after face lifting with the SMAS technique. Despite the declaration of attempts to determine the location and extent of the inflammatory process, there is lack of the specific data on this subject in the article (the temperature distribution is assessed and inflammation is not assessed). In order to monitor the recovery process, the patient had face thermograms examined before the procedure and up to 6 weeks after it, it is a pity that the tests concerned only one person. Such a small sample does not prove that the conclusions are repeatable. According to the authors, the usefulness of thermography in the assessment of post-operative convalescence in facial plastic surgery has a potential in the context of a diagnostic assessment of the dynamics of changes in the healing process. However, as it was already mentioned, one research object does not confirm the rule and the conclusions made. The work is well organized with the introduction, the theoretical part, the application part and conclusions. After the theoretical part, the study presents the results of the research and correctly compiled their analysis. Each part is correctly represented. The research results are correlated with the rest of the article.
Recommendations for the improvement of the manuscript:
- The aim of the work stated in the abstract is interesting, but as it was already mentioned, it was confirmed by the results for only one patient.
- The temperature measurement results are not recorded correctly, in accordance with the applicable standards - different resolution of the recording of the measurement result and SD (eg lines 29-30 and lines 215-219).
- There is no information about the parameters of the used thermographic camera.
- No drawings are displayed in the delivered version of the article.
- Information on the research methodology is not very detailed - was the specified temperature and humidity the same as for the examined patient, how long was the patient in the room before the examination, etc.?
- Did the authors pay attention to other factors disturbing the thermography measurement (eg reflected temperature, transmittance of the atmosphere)?
- Why was the emissivity selected at the level of 0.98?
- Have the authors developed an uncertainty budget to assess the contribution of disturbing factors to thermographic temperature measurement?
- On the basis of what relationships were Tmean and sd calculated?
- Is it correct to declare the resolution of the thermal imaging measurement below 0.1 ° C (line 430)?
Author Response
We would like to thank you very much for your valuable comments. We have made changes taking into account the reviewer's suggestion. All changes in the text are marked in red.
The photos are included in the work, not as additional material as before.
Reviewer comments
The article presents the use of IR thermography in the evaluation of postoperative regeneration processes. The article examines the use of thermography as a method supporting clinical evaluation of the patient's convalescence after face lifting with the SMAS technique. Despite the declaration of attempts to determine the location and extent of the inflammatory process, there is lack of the specific data on this subject in the article (the temperature distribution is assessed and inflammation is not assessed). In order to monitor the recovery process, the patient had face thermograms examined before the procedure and up to 6 weeks after it, it is a pity that the tests concerned only one person. Such a small sample does not prove that the conclusions are repeatable. According to the authors, the usefulness of thermography in the assessment of post-operative convalescence in facial plastic surgery has a potential in the context of a diagnostic assessment of the dynamics of changes in the healing process. However, as it was already mentioned, one research object does not confirm the rule and the conclusions made. The work is well organized with the introduction, the theoretical part, the application part and conclusions. After the theoretical part, the study presents the results of the research and correctly compiled their analysis. Each part is correctly represented. The research results are correlated with the rest of the article.
Recommendations for the improvement of the manuscript:
The aim of the work stated in the abstract is interesting, but as it was already mentioned, it was confirmed by the results for only one patient.
The temperature measurement results are not recorded correctly, in accordance with the applicable standards - different resolution of the recording of the measurement result and SD (eg lines 29-30 and lines 215-219).
Thank you very much for the important remarks of the reviewer. The resolution of the temperature results for all areas in with the SD was calculated have been were standardized to decimals for Tmean and to hundredths for SD (both in the text and Tables) . The error in writing the numerical values ​​was due to the lack of zero values
For tympanic and MPC value it is impossible, because the measurement is in those case is unitary (not an average of the area).
There is no information about the parameters of the used thermographic camera.
Details regarding the technical parameters of the camera have been added
No drawings are displayed in the delivered version of the article.
Due to the necessity to adjust the layout of the photos to the entire content of the manuscript, the photos were placed in separate files to make it easier for the editors to arrange them. According to reviewer suggestion they were also pasted in the current file.
Information on the research methodology is not very detailed - was the specified temperature and humidity the same as for the examined patient, how long was the patient in the room before the examination, etc.?
All the necessary information confirming the compliance of the tests with the standards are included in the text:
All measurements were performed in accordance with the European Association of Thermology standards, in a room with a surface area of 12 m2, with no heat radiators and minimum solar irradiation. In the measurement location, humidity was 55-60%, and temperature was 25℃ ± 0.5. Patient first entered the room for the recommended 20-minute acclimation, to stabilize heat exchange between the body and its surroundings prior to thermal imaging. The emission coefficient of the skin was set at 0.98. Images were captured at a distance of 1.5 m, with study subject standing up (Fujimasa 1995, Å»uber and Jung 1997, Bauer and DereÅ„ 2014). To minimize the impact of daily rhythms on body temperature fluctuations, all imaging was performed at the same time- in the morning (before 11 a.m.). All the necceseary information
Did the authors pay attention to other factors disturbing the thermography measurement (eg reflected temperature, transmittance of the atmosphere)?
The total radiation received by the camera (Wtot) comes from three sources: the emission of the target object (Eobj), the emission of the surroundings and reflected by the object (Erefl) and the emission of the atmosphere (Eatm). The transmittance of the atmosphere is generally estimated using the distance from the object to the camera and the relative humidity. In general, this value is very close to one. The temperature of the atmosphere is obtained using a common thermometer. However, as the emittance of the atmosphere is very close to zero (1 – τatm), this parameter has little influence on the temperature measurement. The most important calibration parameter for temperature measurement using IRT is emissivity. This parameter indicates how much radiation is emitted from the target object compared to that from a blackbody at the same temperature. The place where temperature measurement is carried out must be a room at controlled homogeneous temperature and free from any secondary infrared sources, such as lamps. The subjects require an acclimation time in the room to achieve thermal equilibrium. They must rest during this time in a comfortable position. Prior to the measurements, subjects must follow some instructions, such as no sunbathing and no use of lotions or creams. Patients also must abstain from consuming alcohol or caffeine for a 4-h period prior to the start of the procedure. Many different applications can be found where these procedures are followed
[Usamentiaga, et al. Infrared Thermography for Temperature Measurement and Non-Destructive Testing. Sensors (Basel). 2014 Jul; 14(7): 12305–12348. doi: 10.3390/s140712305]
Why was the emissivity selected at the level of 0.98?
This level of emissivity was set knowing that the most frequently reported values 0.985 or 0.980 correspond to the skin in ‘normal conditions’, that means clean skin without any injuries.
[Charlton, et all. The effect of constitutive pigmentation on the measured emissivity of human skin. PLoS One. 2020; 15(11): e0241843; Bernard et all. Infrared camera assessment of skin surface temperature – Effect of emissivity. Physica Medica 2013, 29,6: 583-591; Dell’Isola et all. Noncontact Body Temperature Measurement: Uncertainty Evaluation and Screening Decision Rule to Prevent the Spread of COVID-19. Sensors 2021, 21, 346].
Have the authors developed an uncertainty budget to assess the contribution of disturbing factors to thermographic temperature measurement?
On the basis of what relationships were Tmean and sd calculated?
Temperature values ​​from the areas (ROI): Tmean and SD were determined using the software FLIR ResearchIR 4, USA which allows to record the maximum, minimum and average temperature as well as Sd from a ROI each time.
Is it correct to declare the resolution of the thermal imaging measurement below 0.1 ° C (line 430)?
Thank you very much to the reviewer for his valuable attention. This was an editorial error, correct value has been corrected into: .. with a resolution of less than ±1°C.

Round 2
Reviewer 2 Report
Please add the following information in the text: Temperature values ​​from the areas (ROI): Tmean and SD were determined using the software FLIR ResearchIR 4, USA which allows to record the maximum, minimum and average temperature as well as Sd from a ROI each time.